# Enterohaemorrhagic *E. coli* utilizes host- and microbiota-derived L-malate as a signaling molecule for intestinal colonization

Bin Liu[1,2,3], Lingyan Jiang [1,2,3], Yutao Liu [1,2,3], Hongmin Sun[1,2], Jun Yan[1,2], Chenbo Kang[1,2] & Bin Yang [1,2] ✉

The mammalian gastrointestinal tract is a complex environment that hosts a diverse microbial community. To establish infection, bacterial pathogens must be able to compete with the indigenous microbiota for nutrients, as well as sense the host environment and modulate the expression of genes essential for colonization and virulence. Here, we found that enterohemorrhagic *Escherichia coli* (EHEC) O157:H7 imports host- and microbiota-derived L-malate using the DcuABC transporters and converts these substrates into fumarate to fuel anaerobic fumarate respiration during infection, thereby promoting its colonization of the host intestine. Moreover, L-malate is important not only for nutrient metabolism but also as a signaling molecule that activates virulence gene expression in EHEC O157:H7. The complete virulence-regulating pathway was elucidated; the DcuS/DcuR two-component system senses high L-malate levels and transduces the signal to the master virulence regulator Ler, which in turn activates locus of enterocyte effacement (LEE) genes to promote EHEC O157:H7 adherence to epithelial cells of the large intestine. Disruption of this virulence-regulating pathway by deleting either *dcuS* or *dcuR* significantly reduced colonization by EHEC O157:H7 in the infant rabbit intestinal tract; therefore, targeting these genes and altering physiological aspects of the intestinal environment may offer alternatives for EHEC infection treatment.

Enterohemorrhagic *Escherichia coli* (EHEC) O157:H7 is a worldwide food-borne pathogen that typically causes diarrhea, hemorrhagic colitis or lethal hemolytic uremic syndrome (HUS)[1,2]. EHEC O157:H7 specifically colonizes the human large intestine and causes lesions on intestinal epithelial cells, termed attaching and effacing (AE) lesions, which are characterized by the destruction of microvilli, the intimate adherence of bacteria to the intestinal epithelial cells and rearrangement of the cytoskeleton to form a pedestal-like structure cupping individual bacterial cells[1,3]. The genes involved in the formation of AE lesions are encoded within a chromosomal

pathogenicity island named the locus of enterocyte effacement (LEE), which is mainly organized into the following five polycistronic operons: LEE1, LEE2, LEE3, LEE4 and LEE5[4]. The first gene of LEE1 encodes the master LEE regulator Ler, which activates the transcription of these genes from LEE1 to LEE5[4,5]. The other genes located in the LEE pathogenicity island mainly encode a type III secretion system (T3SS) that exports effector molecules; the outer membrane adhesin intimin and its translocated receptor Tir, which are necessary for intimate attachment to host epithelial cells; and several secreted proteins as a part of the T3SS, which are important for the

[1]TEDA Institute of Biological Sciences and Biotechnology, Nankai University, TEDA, Tianjin 300457, P. R. China. [2]The Key Laboratory of Molecular Microbiology and Technology, Ministry of Education, Tianjin 300071, P. R. China. [3]These authors contributed equally: Bin Liu, Lingyan Jiang, Yutao Liu. ✉e-mail: yangbin@nankai.edu.cn

modification of host cell signal transduction during the formation of AE lesions[1,4].

The mammalian gastrointestinal tract is populated by a dense and highly adapted microbiota that poses substantial challenges to enteric pathogens[6]. To successfully cause disease, EHEC O157:H7 has evolved multiple mechanisms to compete with the indigenous microbiota for nutrients, as well as sense a variety of host intestinal signals and modulate the expression of its virulence genes[7,8]. One mechanism by which EHEC O157:H7 can compete with the microbiota is by using a distinct metabolic repertoire. EHEC is able to utilize ethanolamine, galactose, hexuronate, mannose and ribose as nitrogen or carbon sources, while commensal *E. coli* cannot use such nutrients[7,9–11]. In addition to competing for nutrients and proliferating to sufficient levels, EHEC O157:H7 must precisely regulate virulence gene expression at the appropriate time and place during infection while reducing metabolic cost and/or alerting the host immune system[8,12–15]. EHEC O157:H7 senses various host- or microbiota-derived metabolites to recognize the host intestinal environment and coordinates the expression of virulence genes accordingly. Ethanolamine, epinephrine, norepinephrine, bicarbonate, butyrate, cysteine and vitamin B12 have been shown to promote EHEC O157:H7 adherence to host cells and/or virulence gene expression[12,16–18]. In contrast, biotin, fucose and D-serine repress EHEC O157:H7 virulence[12,15,18,19]. However, the overall regulatory networks and underlying mechanisms by which host- or gut microbiota-produced metabolites regulate EHEC O157:H7 growth and virulence remain largely unknown.

C4-dicarboxylates such as L-malate, L-aspartate, fumarate and succinate support the aerobic and anaerobic growth of *Escherichia coli* and other enteric bacteria[20–24]. In aerobic metabolism, C4-dicarboxylates are mainly taken up by DctA and are oxidized to $CO_2$ by the tricarboxylic acid (TCA) cycle with pyruvate bypass[24–26]. In anaerobic metabolism, C4-dicarboxylates are mainly taken up by three anaerobic C4-dicarboxylate/succinate antiporters (DcuA, DcuB and DcuC)[27–29]. After uptake, L-malate and L-aspartate are first converted to fumarate by fumarase FumB and aspartase AspA, respectively; fumarate is then reduced to succinate during fumarate respiration using the fumarate reductase FrdABCD[22,23,25,26]. The succinate produced cannot be oxidized because the TCA cycle is nonfunctional, and it is therefore excreted[25,26,30]. The significance of anaerobic fumarate respiration with C4-dicarboxylates, such as L-malate and L-aspartate, has been confirmed for growth and colonization in the mammalian intestinal tract by commensal and pathogenic bacteria. *E. coli* mutants with defective anaerobic fumarate respiration genes (*dcuSR*, *frdA*, *dcuB*, *dcuA* and *aspA*) were impaired in terms of their ability to colonize the intestine in the streptomycin-treated mouse model[22,31]. A recent work demonstrated that *Salmonella* Typhimurium imports L-malate and L-aspartate using the DcuABC transporters and converts these substrates into fumarate to fuel anaerobic $H_2$/fumarate respiration after arrival in the microbiota-colonized murine gut; therefore, *Salmonella* can grow in the face of an intact gut microbiota to establish infection[21]. However, whether fumarate respiration contributes to intestinal colonization by EHEC O157:H7 remains unclear.

The utilization of C4-dicarboxylates (L-malate, L-aspartate, fumarate and succinate) is regulated at the transcription level by the DcuSR two-component system (TCS), which is able to sense the presence of external C4-dicarboxylates and induces the expression of genes required for C4-dicarboxylate metabolism, including fumarate respiration[32–34]. The approximate $K_m$ for the induction of the DcuSR-dependent genes by C4-dicarboxylates is in the micromolar range, and L-malate is the preferred substrate of the sensor DcuS[34,35]. However, whether EHEC O157:H7 utilizes C4-dicarboxylates as signaling molecules to mediate virulence gene expression in the large intestine remains unknown.

In the current study, we demonstrated that EHEC O157:H7 utilizes L-malate as an important nutrient to drive fumarate respiration during infection, thereby promoting its growth and colonization in the host intestine. Moreover, EHEC O157:H7 responds to L-malate as a signaling molecule to activate virulence gene expression. Further investigation revealed the complete signal transduction pathway, wherein the DcuSR TCS senses L-malate as a signal to directly activate the expression of *ler* and other LEE genes. The presence of this regulatory pathway was also verified in an additional 7 EHEC strains from different serotypes. Blocking this pathway by deleting either *dcuS* or *dcuR* significantly reduced EHEC O157 virulence both in vitro and in vivo; therefore, these genes might be targeted in the future for the development of effective prevention and therapeutic strategies against infections caused by this virulent pathogen.

## Results

### The large intestine is the optimal colonization site for EHEC O157:H7 in infant rabbits

Although different mouse infection models that mirror various aspects of EHEC O157:H7 pathogenesis or disease have been developed, many require that mice undergo artificial manipulations, such as prolonged dietary restriction to promote colonization, mitomycin C injection to facilitate the expression of Shiga toxin, or antibiotic treatment to reduce the abundance of the normal flora that can inhibit the establishment of an exogenous infection[36]. Furthermore, the mice do not develop diarrhea, colitis, or AE lesions following EHEC O157:H7 inoculation[37]. Infant rabbits provide a more readily available animal model without the requirement for additional treatments, wherein EHEC O157:H7 colonizes the infant rabbit intestine, leading to the formation of AE lesions and the development of severe diarrhea and intestinal inflammation[38,39]. Therefore, this animal model was chosen in this study to compare the colonization of EHEC O157:H7 in the small and large intestines. The EHEC O157:H7 loads in the mid-colon (representative of the large intestine) were much higher than those in the mid-ileum (representative of the small intestine) during the 7-day course of infection (Fig. 1a). In both the ileum and colon, the number of EHEC O157:H7 cells recovered from the intestinal tissues increased from 1 d to 4 d and then leveled off for the remainder of the study (Fig. 1a). These results indicated that the colon is the optimal colonization site for EHEC O157:H7 in infant rabbits, which supports the idea that EHEC O157:H7 is a large-intestinal pathogen in humans[40].

### L-malate is highly abundant in the colons of infant rabbits

To further explore whether specific compounds in the colonic fluid promote intestinal colonization by EHEC O157:H7, we compared metabolites in the ileal and colonic contents using untargeted metabolomics analysis. Our metabolomics results showed that the metabolic landscape in these two types of intestinal contents was quite diverse (Fig. 1b and Supplementary Data 1). Several amino acids (including L-lysine, L-arginine, L-threonine, L-glutamine and sarcosine), two carbohydrates and carbohydrate conjugates (D-threitol and raffinose), taurolithocholic acid, adynerin, galactonic acid, and indoxyl sulfate were present at greater abundance in the ileum (Fig. 1b and Supplementary Data 1). In contrast, the levels of four organic acids and their derivatives (L-malate, ketoisocaproic acid, N-acetyl-L-glutamate and acamprosate), five lipids and lipid-like molecules (hydroxyisocaproic acid, pentadecanoic acid, lithocholic acid, deoxycholic acid and myristic acid), three nucleotides and nucleotide analogs (uridine, inosine and pseudouridine), three organoheterocyclic compounds (nicotinate, hypoxanthine and 4-pyridoxic acid), three carbohydrates and carbohydrate conjugates (D-mannose 1-phosphate, D-lyxose and L-arabinose), two dipeptides (D-alanyl-D-alanine and gamma-L-glutamyl-L-valine), two benzenoids (gentisic acid and phenylpyruvate), guconolactone, DL-3-phenyllactic acid, 1-palmitoyl-2-oleoyl-phosphatidylglycerol and scytalone were significantly elevated in the colon compared with the ileum (Fig. 1b and Supplementary Data 1). Of particular interest, the level of L-malate varied the most among these metabolites, and its concentration in the colon was 34.7-fold higher than that in the ileum (Fig. 1b and

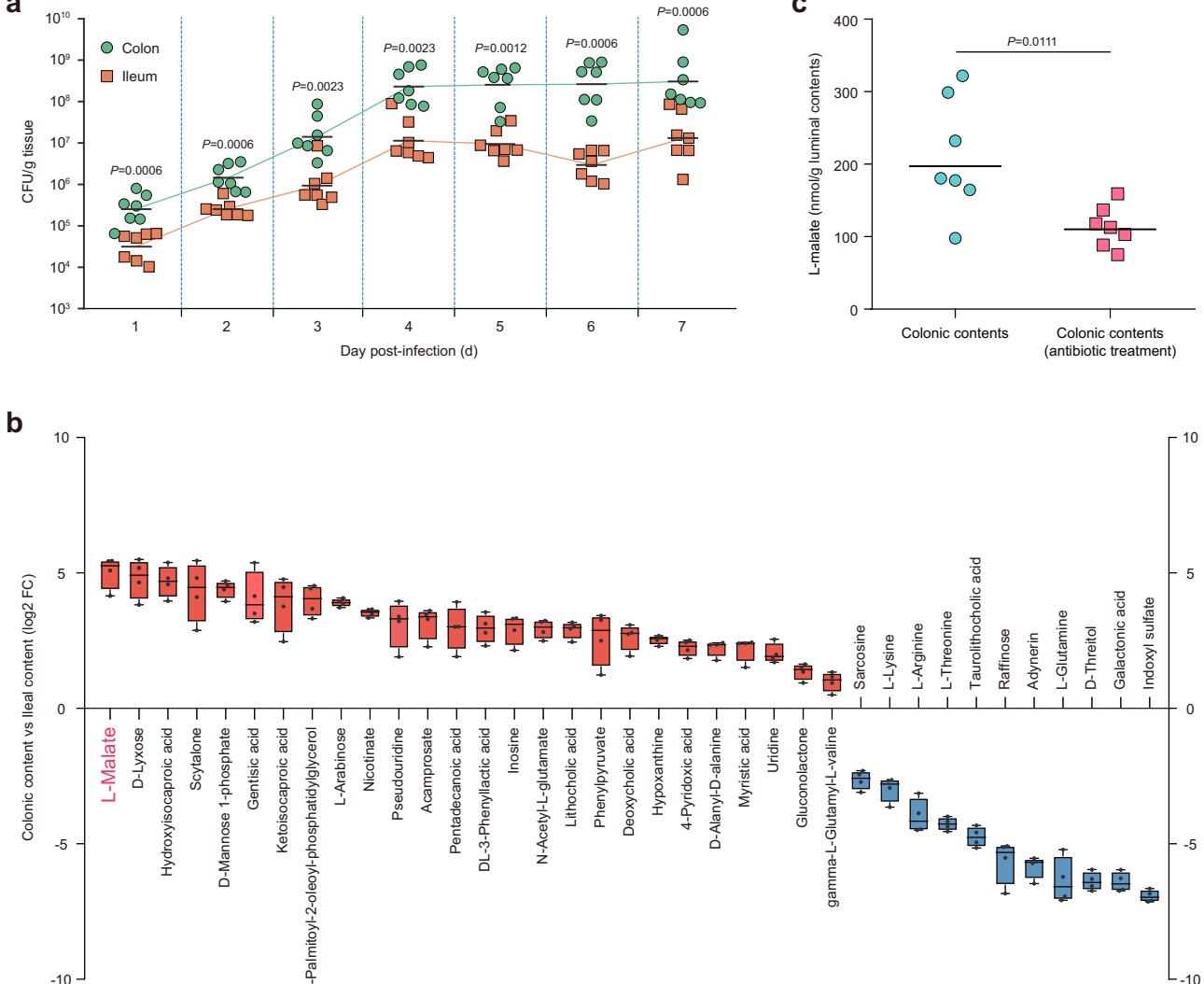

**Fig. 1 | The L-malate-rich large intestine is the predominant site of EHEC O157:H7 colonization in infant rabbits. a** Bacterial counts of wild-type EHEC O157:H7 (abbreviated as O157 WT in all figures) recovered from the ileum and colon of infant rabbits at 1-7 d post-infection. **b** Relative abundance of metabolites that were differentially abundant in the colon versus that in the ileum (blue and red: higher levels in ileal and colonic contents, respectively). Boxes represent the interquartile range, with the vertical line representing the median value. Error bars represent maximum and minimum values (*n* = 4 biologically independent samples). **c** Quantification of L-malate concentrations in the colonic contents obtained from conventional and antibiotic-treated infant rabbits. For (**a, c**), the horizontal lines represent the geometric means. *n* = 7 infant rabbits per group were used. Statistical significance was assessed via the two-sided Mann–Whitney rank-sum test. Source data are included in Source Data file.

Supplementary Data 1). The L-malate level in the colon of infant rabbits was further quantified using an L-Malate Colorimetric Assay Kit, and the concentration was approximately 200 nmol/g luminal contents (Fig. 1c). Assuming that particulate matter and cells occupy 90% of the colonic lumen, the L-malate concentration in the extracellular space was estimated to be 1-3 mM. Elimination of the microbiota by antibiotic treatment significantly reduced the concentration of L-malate in the colon of infant rabbits by approximately 2-fold (110 nmol/g luminal contents), suggesting that L-malate originates from both the host and the microbiota (Fig. 1c). The large amount of L-malate present in the colonic lumen suggested that L-malate may be utilized by EHEC O157:H7 as a nutrient or a virulence gene regulatory signal for colonization in the host large intestine.

## EHEC O157:H7 utilizes L-malate as an important nutrient for fumarate respiration during colonization of the large intestine

To further investigate whether EHEC O157:H7 utilizes L-malate as a nutrient source to benefit growth during intestinal colonization,

mutants defective in L-malate transport or metabolism were constructed and used to evaluate the intestinal colonization capacity in the infant rabbit intestine. During competitive infection of infant rabbits, the Δ*dcuA* and Δ*dcuC* mutants exhibited a mild intestinal colonization defect (mean CI = 0.32 and 0.47, respectively; Fig. 2a). In contrast, the Δ*dcuB* mutant exhibited a more severe colonization defect (mean CI = 0.03; Fig. 2a), indicating that uptake of L-malate is mainly performed by DcuB during colonization by EHEC O157:H7 in the infant rabbit intestine. The colonization defects of the Δ*dcuA*, Δ*dcuB* and Δ*dcuC* mutants could be restored by complementation with the corresponding functional genes (Fig. 2a). Considering that DcuA, DcuB and DcuC can functionally complement each other to support anaerobic fumarate-dependent growth on C4-dicarboxylates in vitro[41], genetic redundancy might mask the fitness consequences of individual transporter mutants. Therefore, we next assessed the colonization phenotype of the Δ*dcuABC* triple mutant. Indeed, the Δ*dcuABC* triple mutant exhibited an approximately 90-fold attenuation of intestinal colonization compared with wild-type EHEC O157:H7 (mean CI = 0.011;

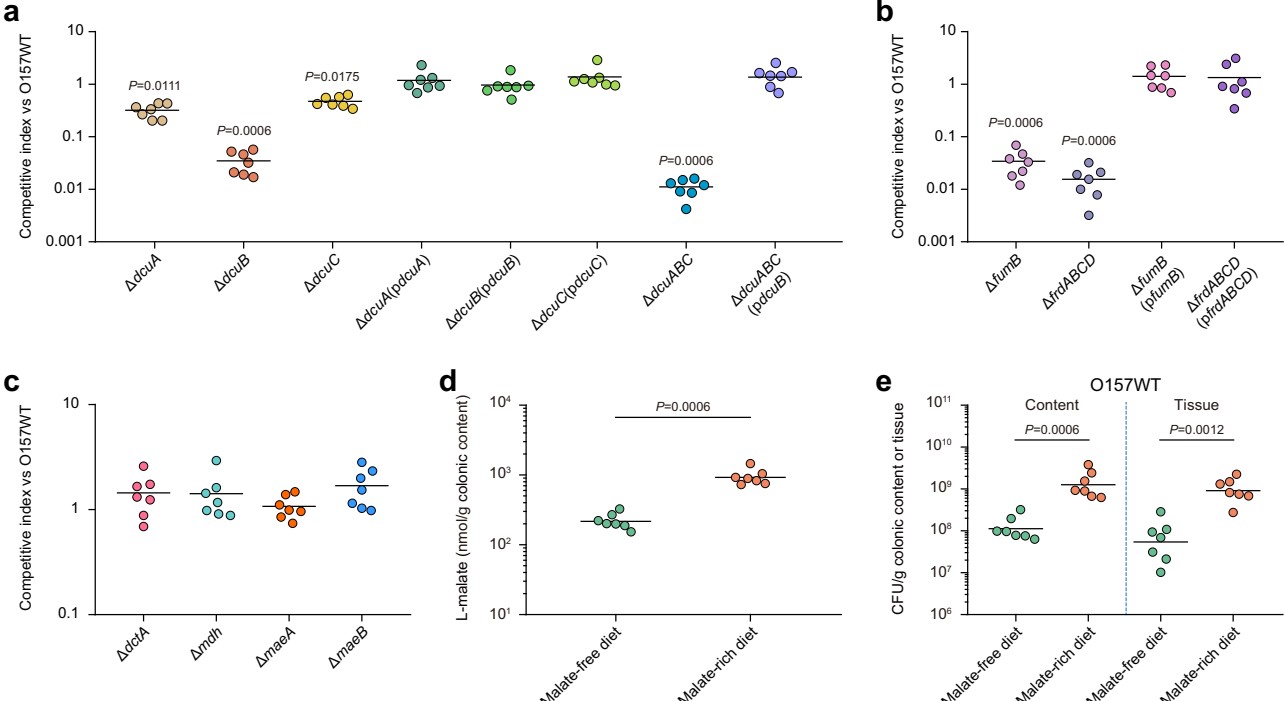

**Fig. 2 | EHEC O157:H7 utilizes L-malate as an important nutrient for fumarate respiration during intestinal colonization. a–c** Competition index (CI) analysis between O157 WT and its mutant or complemented strains. Groups of infant rabbits were intragastrically inoculated with a 1:1 ratio of the O157 WT and its mutant or complemented strains. Bacteria were recovered from colonic contents 4 d post-inoculation. Competitive indexes were calculated using the relative abundance of each strain in the colonic contents, corrected by the ratio in the inoculum. **d** Quantification of L-malate concentrations in the colonic contents obtained from infant rabbits fed an L-malate-free diet or an L-malate-rich diet. **e** Bacterial counts recovered from colonic contents and tissues of L-malate-free diet-fed or L-malate-rich diet-fed infant rabbits infected with O157 WT. The horizontal lines indicate the geometric mean for each group. *n* = 7 infant rabbits per group were used. Statistical significance was assessed via the Kruskal–Wallis test with Dunn's *post hoc* test (**a**–**c**) or the two-sided Mann–Whitney rank-sum test (**d**, **e**). Source data are included in Source Data file.

Fig. 2a). The colonization defects of the Δ*dcuABC* triple mutant could be fully complemented by the presence of a plasmid expressing *dcuB* (Fig. 2a), further confirming that DcuB is the main anaerobic L-malate transporter.

In addition to anaerobic L-malate transport, deletion of genes encoding enzymes for anaerobic fumarate respiration also significantly affected intestinal colonization by EHEC O157:H7. In competitive infection assays, colonization by the Δ*fumB* mutant, which is unable to convert L-malate to fumarate, and the Δ*frdABCD* mutant, which is unable to convert fumarate to succinate, was significantly attenuated compared with colonization by wild-type EHEC O157:H7 (Fig. 2b and Supplementary Fig. 1). The colonization defects of the Δ*fumB* and Δ*frdABCD* mutants were fully complemented by the presence of a plasmid expressing corresponding functional genes (Fig. 2b). Furthermore, the Δ*frdABCD*Δ*dcuABC* mutant exhibited similar intestinal colonization as the Δ*frdABCD* mutant (mean CI = 1.16; Supplementary Fig. 2), indicating that fumarate and its precursors (such as L-malate) internalized through the DcuABC transporter are eventually catabolized by fumarate respiration using FrdABCD. In contrast, deletion of genes involved in aerobic L-malate transport and metabolism, including *dctA* (encoding aerobic L-malate transport), *mdh* (encoding NAD-dependent malate dehydrogenase), and *maeA* and *maeB* (encoding NAD(P)-dependent malic enzyme), had no obvious effects on colonization by EHEC O157:H7 in the intestines of infant rabbits (Fig. 2c and Supplementary Fig. 1).

The effect of the L-malate status in the infant rabbit intestinal tract on colonization by EHEC O157:H7 was further investigated using infant rabbits given an L-malate-free diet (sterilized water containing 10% lactose) or an L-malate-rich diet (sterilized water containing 10% lactose and 10 mM L-malate) for 2 d. Feeding infant rabbits an L-malate-

rich diet resulted in an approximately 4.3-fold increase in L-malate levels in the colonic contents in comparison with the levels observed under an L-malate-free diet (Fig. 2d). Accordingly, infant rabbits maintained on an L-malate-rich diet presented with an increased EHEC O157:H7 load in both the colonic contents and tissues compared with L-malate-free diet-fed infant rabbits 4 d postinfection (Fig. 2e). Therefore, colonization by EHEC O157:H7 in the large intestine was enhanced by increasing L-malate levels, based on feeding infant rabbits a high-L-malate diet. Collectively, these results indicated that EHEC O157:H7 utilizes L-malate as an important nutrient to drive fumarate respiration, thereby promoting EHEC O157:H7 survival and colonization in the host large intestine.

## Genes involved in anaerobic L-malate metabolism are upregulated in a DcuSR-dependent manner during EHEC O157:H7 colonization in infant rabbit intestines

The expression of genes encoding L-malate transporters and enzymes that catalyze L-malate metabolism was subsequently analyzed during EHEC O157:H7 colonization in the large intestine of infant rabbits. The transcription of genes essential for anaerobic L-malate transport and fumarate respiration, including *dcuB*, *dcuC*, *fumB* and *frdA*, was significantly upregulated during EHEC O157:H7 colonization in the large intestine of infant rabbits compared to that during bacterial cell growth in Dulbecco's modified Eagle medium (DMEM) (Fig. 3a). In contrast, the expression of *dcuA* and other genes encoding aerobic L-malate-metabolizing enzymes (including *mdh*, *maeB*, *maeA*, and *sdhC*) was either unchanged or downregulated during EHEC O157:H7 colonization in the large intestine of infant rabbits (Fig. 3a, b). The upregulation of *dcuB*, *fumB* and *frdA* with EHEC O157:H7 wild-type colonization in the large intestine of infant rabbits was completely

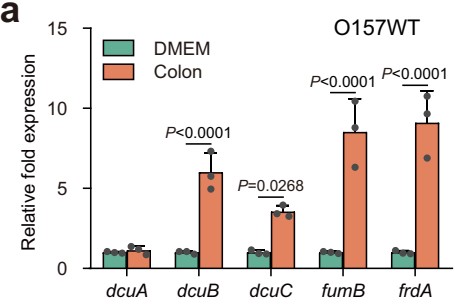

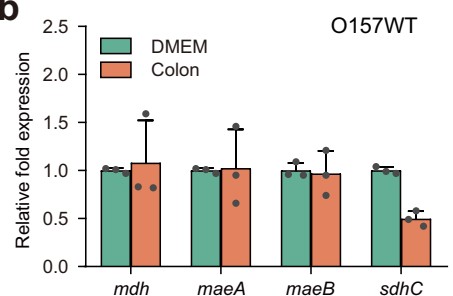

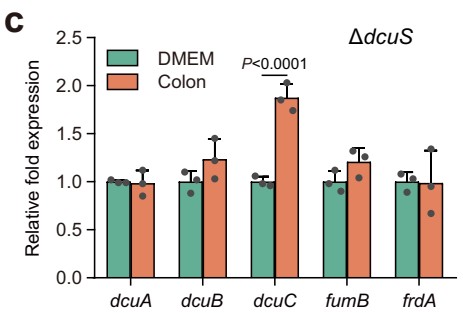

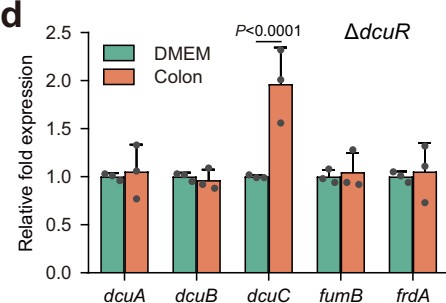

**Fig. 3 | Genes involved in anaerobic L-malate metabolism are upregulated in a DcuSR-dependent manner during EHEC O157:H7 colonization in the infant rabbit intestine. a** qRT–PCR was performed to measure the expression of *dcuA*, *dcuB*, *dcuC*, *fumB* and *frdA* in O157 WT cells grown in DMEM or recovered from the colons of infected infant rabbits. **b** qRT–PCR was performed to measure the expression of *mdh*, *maeA*, *maeB* and *sdhC* in O157 WT cells grown in DMEM or

recovered from the colons of infected infant rabbits. qRT–PCR was performed to measure the expression of *dcuA*, *dcuB*, *dcuC*, *fumB* and *frdA* in the Δ*dcuS* (**c**) or Δ*dcuR* (**d**) strain grown in DMEM or recovered from the colons of infected infant rabbits. Data are presented as the mean ± SD of three independent biological replicates (*n* = 3). Statistical significance was assessed via two-way ANOVA followed by Sidak's *post hoc* test. Source data are included in Source Data file.

abolished by deletion of *dcuS* or *dcuR* (Fig. 3c, d). Interestingly, the expression of *dcuC* was still induced, albeit to a markedly lower extent, in EHEC O157:H7 in the absence of *dcuS* or *dcuR* (Fig. 3c, d), consistent with previous findings showing that *dcuC* is not a target of DcuSR regulation[33]. Collectively, these results indicated that the genes associated with anaerobic L-malate transport and metabolism were significantly upregulated during EHEC O157:H7 colonization in the large intestine of infant rabbits, and this regulation was mediated by the DcuSR TCS.

### L-malate enhances EHEC O157:H7 adherence to host epithelial cells by promoting LEE gene expression

As our previous results indicated that EHEC O157:H7 exploits L-malate as a nutrient source to promote colonization in the infant rabbit intestine, we next aimed to determine whether EHEC O157:H7 utilizes L-malate as a signaling molecule to mediate bacterial adherence and virulence gene expression. The ability of EHEC O157:H7 to adhere to HeLa cells was significantly increased by adding 150 μM L-malate to DMEM and further increased in the presence of 300 μM, 500 μM, 1000 μM and 3000 μM L-malate (Fig. 4a). In contrast, less than 100 μM L-malate had no obvious effect on EHEC O157:H7 adherence to host cells (Fig. 4a). A similar increase was also observed during infection of Caco-2 intestinal epithelial cells (Fig. 4b). The growth of EHEC O157:H7 in DMEM was not affected by the presence of 150 μM and 500 μM L-malate (Fig. 4c), indicating that the positive effect of L-malate on EHEC O157:H7 adherence was not due to a difference in growth rates. Fluorescein actin staining (FAS) assays revealed that EHEC O157:H7 formed more pedestals, which are characteristic of AE lesions, on HeLa cells in the presence of 150 μM L-malate (Fig. 4d). The percentage of HeLa cells infected, as estimated from 300 cells per strain, increased from 37.0 to 63.7% (Fig. 4d), with the number of pedestals per infected cell increasing to 18.8 in the presence of 150 μM L-malate from 10.1 in the absence of L-malate (Fig. 4d). In accordance with the adherence

results, the expression of representative LEE genes, including *ler* (the master regulator of LEE genes), *escT* (LEE1), *escC* (LEE2), *escN* (LEE3), *eae* (intimin, LEE5), *tir* (intimin receptor, LEE 5) and *espB* (LEE 4), was also significantly increased in the presence of 150 μM L-malate in EHEC O157:H7 compared to the expression levels observed without L-malate (Fig. 4e). Additionally, the protein levels of intimin and the translocated intimin receptor Tir were also significantly increased in the presence of 150 μM L-malate based on Western blotting analysis (Fig. 4f). Collectively, these results indicate that EHEC O157:H7 senses L-malate as a signaling molecule to increase its adherence to host epithelial cells by promoting LEE gene expression.

### The promotion of EHEC O157:H7 adherence and LEE gene expression by L-malate is mediated by DcuSR TCS

Whether the DcuSR TCS is involved in the L-malate-induced activation of LEE genes in EHEC O157:H7 was subsequently examined. Deletion of *dcuS* or *dcuR* in EHEC O157:H7 resulted in significant reductions in adherence to HeLa cells and Caco-2 cells (Fig. 5a), AE lesion formation (Fig. 5b), and LEE gene expression (Fig. 5c). These changes could be restored to wild-type levels when a low-copy plasmid carrying the functional gene was introduced into the corresponding mutant strains (Fig. 5a−c). Furthermore, the transcript levels of the LEE genes were also significantly decreased in the Δ*dcuS* mutant and the Δ*dcuR* mutant compared with wild-type EHEC O157:H7 during colonization in the large intestine of infant rabbits (Fig. 5d), indicating that the DcuSR TCS is required for EHEC O157:H7 LEE gene expression and T3SS synthesis in vivo. In contrast, deletion of *dcuABC* had no obvious effects on the adherence of EHEC O157:H7 to HeLa cells and Caco-2 cells (Fig. 5a), indicating that the DcuSR TCS promotion of EHEC O157:H7 adherence was not due to enhanced utilization of L-malate as a nutrient source for fumarate respiration. In addition, all these virulence features of the Δ*dcuS* mutant or the Δ*dcuR* mutant were unaffected by the presence of 150 μM L-malate (Fig. 5e−h). These results revealed that the DcuSR TCS

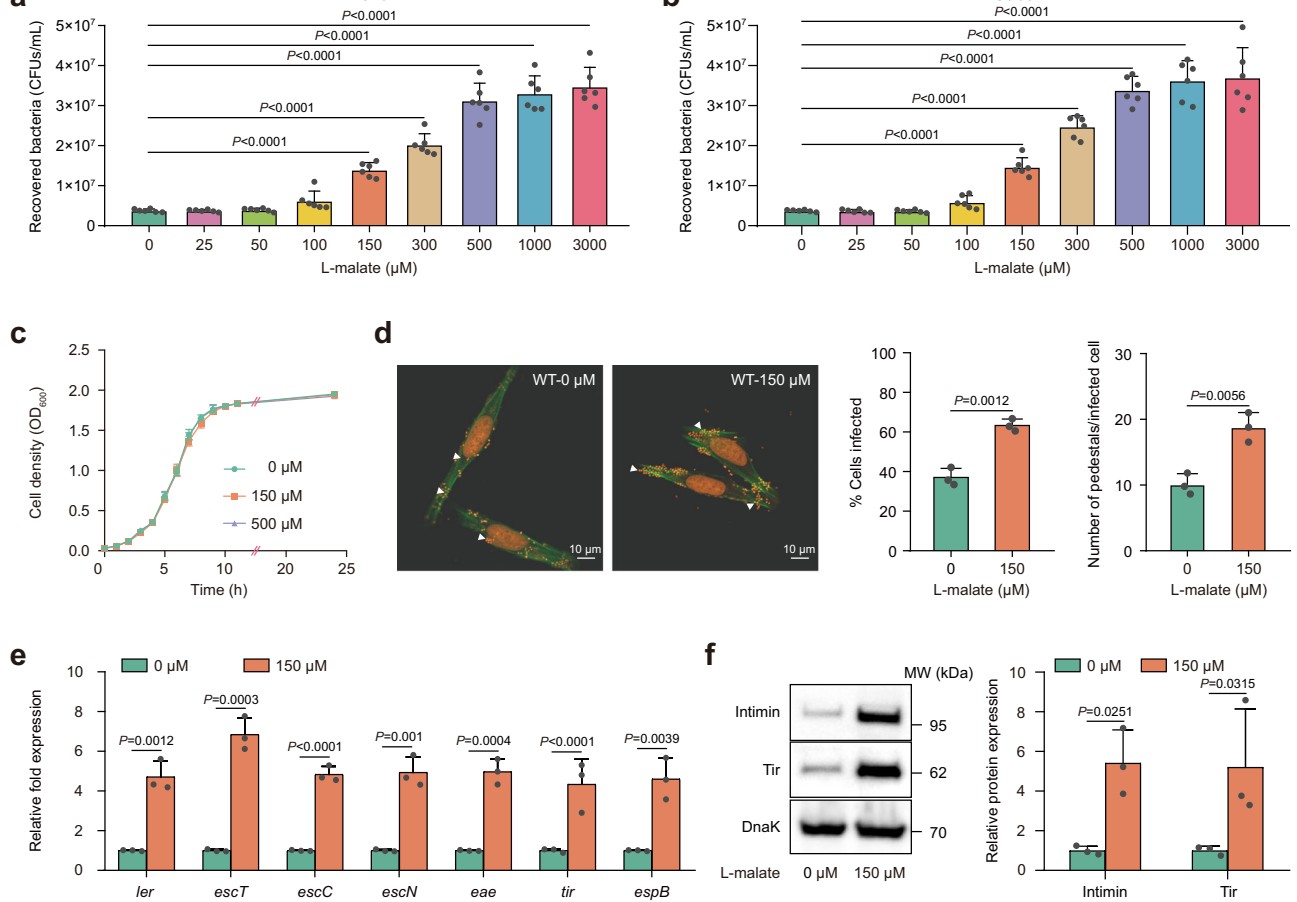

**Fig. 4 | L-malate enhances EHEC O157:H7 adherence to host cells by promoting LEE gene expression.** Adherence of O157 WT to HeLa cells (**a**) or Caco-2 cells (**b**) in DMEM supplemented with different concentrations of L-malate. **c** O157 WT growth under aerobic conditions at 37 °C in DMEM supplemented with different concentrations of L-malate. **d** Fluorescent actin staining (FAS) assay of HeLa cells infected with O157 WT grown in DMEM supplemented with 0 or 150 µM L-malate. HeLa nuclei and bacteria were stained with propidium iodide (red), and the HeLa cell actin cytoskeleton was stained with FITC-phalloidin (green). Pedestals are observed as green punctate structures associated with bacterial cells and are indicated by arrowheads. Original magnification, 63×. Scale bar, 10 µm. The FAS assay results were quantified based on the percentage of HeLa cells infected and the number of pedestals per infected cell after 3 h of incubation (100 HeLa cells per slide, 3 slides each). **e** qRT–PCR to determine LEE gene expression changes in O157 WT grown in DMEM supplemented with 0 or 150 µM L-malate. **f** Representative Western blotting images and quantitative analysis of intimin and its receptor Tir in O157 WT cells grown in DMEM supplemented with 0 or 150 µM L-malate. In (**a**, **b**), data are presented as the mean ± SD of six independent biological replicates ($n = 6$). In (**c**–**f**), data are presented as the mean ± SD of three independent biological replicates ($n = 3$). Statistical significance was assessed via one-way ANOVA followed by Dunnett's post hoc test (**a**, **b**), two-sided Student's $t$ test (**d**) or two-way ANOVA followed by Sidak's post hoc test (**e**, **f**). Source data are included in Source Data file.

positively regulates EHEC O157:H7 virulence features and that the positive effect of L-malate on bacterial adherence and LEE gene expression is mediated by the DcuSR TCS.

## DcuR directly binds to the LEE1 promoter and activates LEE gene expression via Ler

To investigate whether DcuR regulates the expression of LEE genes directly or indirectly, we performed electrophoretic mobility shift assays (EMSAs) and competition assays with purified DcuR-His$_6$ and LEE promoters (P$_{LEE1}$, P$_{LEE2/3}$, P$_{LEE4}$, and P$_{LEE5}$). With increasing concentrations of DcuR, slowly migrating bands were observed for P$_{LEE1}$, and unlabeled P$_{LEE1}$ effectively competed for DcuR binding with FAM-labeled P$_{LEE1}$ (Fig. 6a), indicating that DcuR binds directly and specifically to P$_{LEE1}$ in vitro. Meanwhile, DcuR did not bind to *rpoS* (negative control) or other LEE promoters (P$_{LEE2/3}$, P$_{LEE4}$, and P$_{LEE5}$) under the same experimental conditions (Fig. 6a and Supplementary Fig. 3a). Chromatin immunoprecipitation-quantitative PCR (ChIP–qPCR) analysis further showed that P$_{LEE1}$ was markedly enriched in DcuR-ChIP samples, with levels 8.6-fold higher than those in the mock-ChIP control samples (Fig. 6b), indicating that DcuR was bound to P$_{LEE1}$ in vivo.

In contrast, the enrichment of *rpoS*, P$_{LEE2/3}$, P$_{LEE4}$, and P$_{LEE5}$ was similar in both the DcuR-ChIP and mock-ChIP samples (Fig. 6b and Supplementary Fig. 3b). Using a dye-based DNase I footprinting assay, we further found a specific DcuR-bound sequence containing an 18-base pair motif (5′-TTATCTCACATAATTTAT-3′) (Fig. 6c). This motif is located from positions -62 to -45, counting from the proximal transcriptional start site, which is right upstream of the -35 region (Supplementary Fig. 3c). Deletion of the binding motif (P$_{LEE1}$-1) or mutation of the binding motif to GGCGAGACACGCCGGGCG (P$_{LEE1}$-2) completely abolished the binding of DcuR to P$_{LEE1}$, as determined by EMSAs and competition assays (Fig. 6d), confirming that the TTATCTCACATAATTTAT motif is crucial to the DNA-binding ability of DcuR.

The first LEE1 gene encodes the master LEE regulator Ler, which activates the expression of genes from LEE1 to LEE5[5]. Considering that DcuR can directly bind to P$_{LEE1}$ and positively regulate *ler* expression (Figs. 5c, 6a), we subsequently investigated whether DcuR regulates EHEC O157:H7 adherence and LEE gene expression via Ler. Deletion of *ler* in EHEC O157:H7 markedly reduced bacterial adherence and LEE gene expression (Fig. 6e, f), which is consistent with the function of Ler as a positive regulator of LEE genes. Furthermore, the adherence

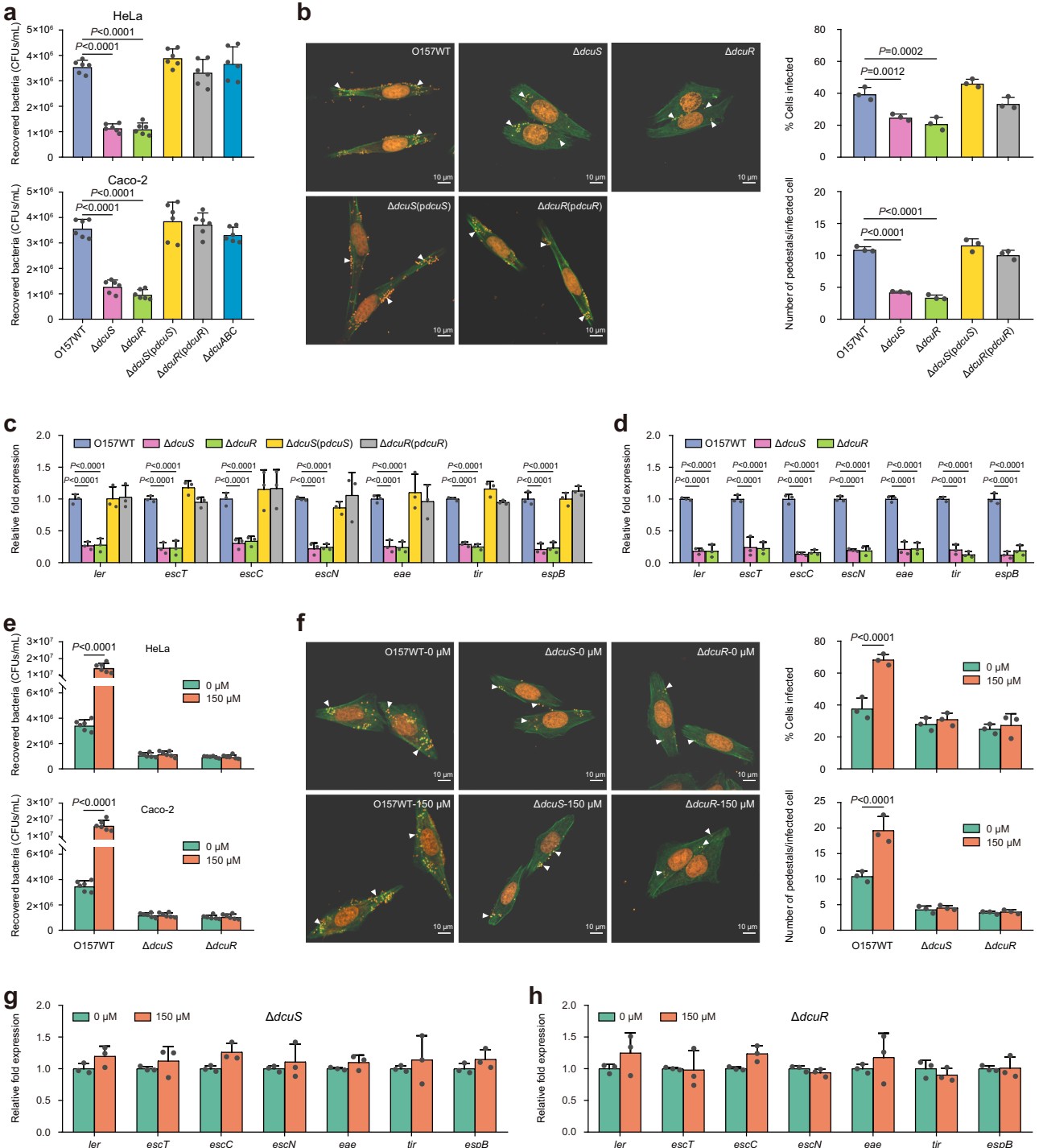

**Fig. 5 | Activation of EHEC O157:H7 adherence and LEE gene expression by L-malate is mediated by the DcuSR two-component system. a** Adherence of O157 WT, Δ*dcuS*, Δ*dcuR*, Δ*dcuS* (p*dcuS*), Δ*dcuR* (p*dcuR*), and Δ*dcuABC* to HeLa cells or Caco-2 cells. **b** FAS assay of HeLa cells infected with O157 WT, Δ*dcuS*, Δ*dcuR*, Δ*dcuS* (p*dcuS*), and Δ*dcuR* (p*dcuR*). **c** qRT–PCR to determine LEE gene expression changes in O157 WT, Δ*dcuS*, Δ*dcuR*, Δ*dcuS* (p*dcuS*), and Δ*dcuR* (p*dcuR*) grown in DMEM. **d** qRT–PCR to determine LEE gene expression changes in O157 WT, Δ*dcuS*, and Δ*dcuR* recovered from the colons of infected infant rabbits. **e** Adherence of O157 WT, Δ*dcuS* and Δ*dcuR* to HeLa cells or Caco-2 cells in DMEM supplemented with 0 or 150 μM L-malate. **f** FAS assay of HeLa cells infected with O157 WT, Δ*dcuS* and Δ*dcuR* in the presence of 0 or 150 μM L-malate. qRT–PCR to determine LEE

gene expression changes in Δ*dcuS* (**g**) and Δ*dcuR* (**h**) grown in DMEM supplemented with 0 or 150 μM L-malate. O157 WT, EHEC O157:H7 wild-type strain; Δ*dcuS*, *dcuS* mutant; Δ*dcuR*, *dcuR* mutant; Δ*dcuS*(p*dcuS*), *dcuS* mutant complemented with *dcuS*; Δ*dcuR* (p*dcuR*), *dcuR* mutant complemented with *dcuR*; Δ*dcuABC*, *dcuABC* triple mutant. In (**a**, **e**), data are presented as the mean ± SD of six independent biological replicates (*n* = 6). In (**b**, **c**, **d**, **f**, **g**, **h**), data are presented as the mean ± SD of three independent biological replicates (*n* = 3). Statistical significance was assessed via one-way ANOVA followed by Dunnett's post hoc test (**a**, **b**), two-way ANOVA followed by Dunnett's post hoc test (**c**, **d**) or two-way ANOVA followed by Sidak's post hoc test (**e**, **f**). Source data are included in Source Data file.

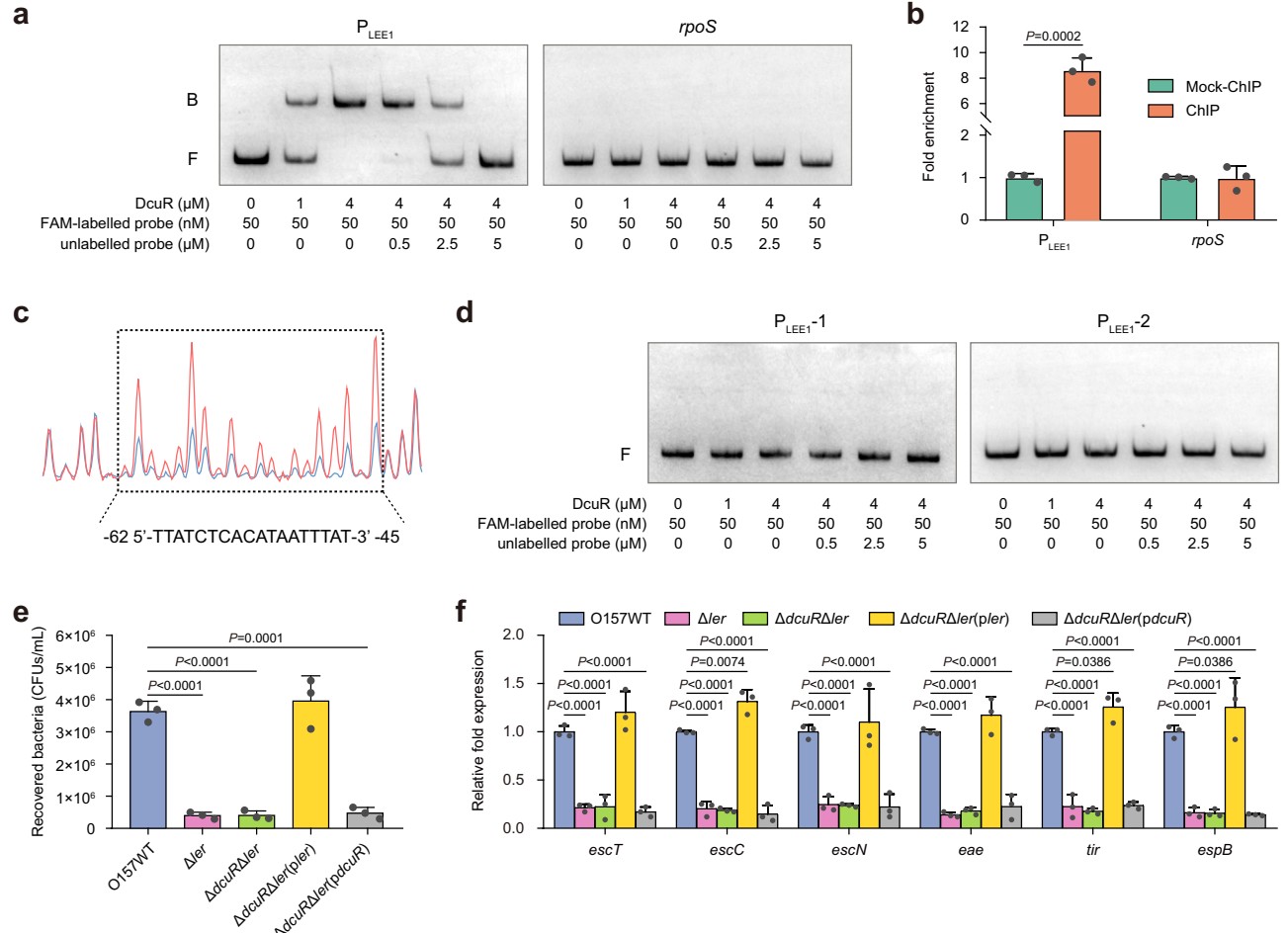

**Fig. 6 | DcuR directly binds to the LEE1 promoter and activates LEE gene expression via Ler. a** Gel mobility shift and competition assays of DcuR with the promoter region of LEE1 and *rpoS* (negative control). Positions of the bound (denoted with "B") and free (denoted with "F") probes are shown on the left, and the concentrations of the probe and purified DcuR are indicated at the bottom of each lane. **b** Fold enrichment of the LEE1 promoter in DcuR-ChIP samples, as measured via ChIP–qPCR. *rpoS* is the negative control. **c** DcuR binds to the motif TTATCT-CACATAATTTAT in the LEE1 promoter region. FAM-labeled probes (40 nM) were used for binding reactions in the absence (red peaks) or presence of 2.5 μM (blue peaks) DcuR. The region protected from DNase I digestion is boxed, and the corresponding nucleotide sequence is shown beneath the electropherogram. **d** Gel mobility shift and competition assays of DcuR with the modified LEE1 promoter region P$_{LEE1}$-1 (without the binding motif) and P$_{LEE1}$-2 (with the mutated motif,

GGCGAGACACGCCGGGCG). **e** Adherence of O157 WT, Δ*ler*, Δ*dcuR*Δ*ler*, Δ*dcuR*-Δ*ler*(p*ler*), and Δ*dcuR*Δ*ler*(p*dcuR*) to HeLa cells. **f** qRT–PCR to determine LEE gene expression changes in O157 WT, Δ*ler*, Δ*dcuR*Δ*ler*, Δ*dcuR*Δ*ler*(p*ler*), and Δ*dcuR*Δ*ler*(p*dcuR*). O157 WT, EHEC O157:H7 wild-type strain; Δ*ler*, *ler* mutant; Δ*dcuR*Δ*ler*, *dcuR/ler* double mutant; Δ*dcuR*Δ*ler*(p*ler*), *dcuR/ler* double mutant complemented with *ler*; Δ*dcuR*Δ*ler*(p*dcuR*), *dcuR/ler* double mutant complemented with *dcuR*. In (**a**, **d**), images are representative of three independent experiments. In (**b**, **e**, **f**), data are presented as the mean ± SD of three independent biological replicates (*n* = 3). Statistical significance was assessed via two-way ANOVA followed by Sidak's post hoc test (**b**), one-way ANOVA followed by Dunnett's post hoc test (**e**) or two-way ANOVA followed by Dunnett's post hoc test (**f**). Source data are included in Source Data file.

capacity and LEE gene expression in the Δ*dcuR*Δ*ler* double mutant were comparable to those in the Δ*ler* single mutant, as evidenced by the adherence assay and qRT–PCR analysis (Fig. 6e, f), indicating that DcuR has no effect on EHEC O157:H7 adherence and LEE expression in the absence of *ler*. Furthermore, complementation of the Δ*dcuR* mutant or the Δ*dcuR*Δ*ler* double mutant with *trc* promoter-controlled *ler* fully restored bacterial adherence capacity and LEE gene expression to the wild-type level in the presence of 0.1 mM IPTG (Fig. 6e, f, and Supplementary Fig. 4a, b), whereas no apparent changes in these virulence features were observed when the Δ*ler* mutant or the Δ*dcuR*Δ*ler* double mutant was complemented with *dcuR* (Fig. 6e, f, and Supplementary Fig. 4a, b). Collectively, these findings indicate that DcuR directly binds to the LEE1 promoter to activate *ler* expression, which in turn activates LEE gene expression to promote EHEC O157:H7 adherence via Ler.

**The positive effect of L-malate on bacterial adherence and LEE gene expression is conserved in EHEC**

To further investigate whether the L-malate signaling regulatory pathway for the control of bacterial adherence and LEE gene expression was also present in other EHEC strains, an additional seven representative EHEC strains with different serotypes were chosen and used to carry out HeLa adherence experiments and qRT–PCR analysis of LEE gene expression in the presence of 0 and 150 μM L-malate. These EHEC strains were first verified through serotyping using polyclonal O-antigen and H-antigen antisera and a multiplex PCR assay to further confirm the presence of intimin and Shiga toxin genes (Supplementary Data 2). Enhanced adherence to HeLa cells and LEE gene expression in the presence of 150 μM L-malate was observed for all seven tested EHEC strains (Fig. 7a–c), indicating that the L-malate signaling regulatory pathway is not EHEC O157-specific but a common

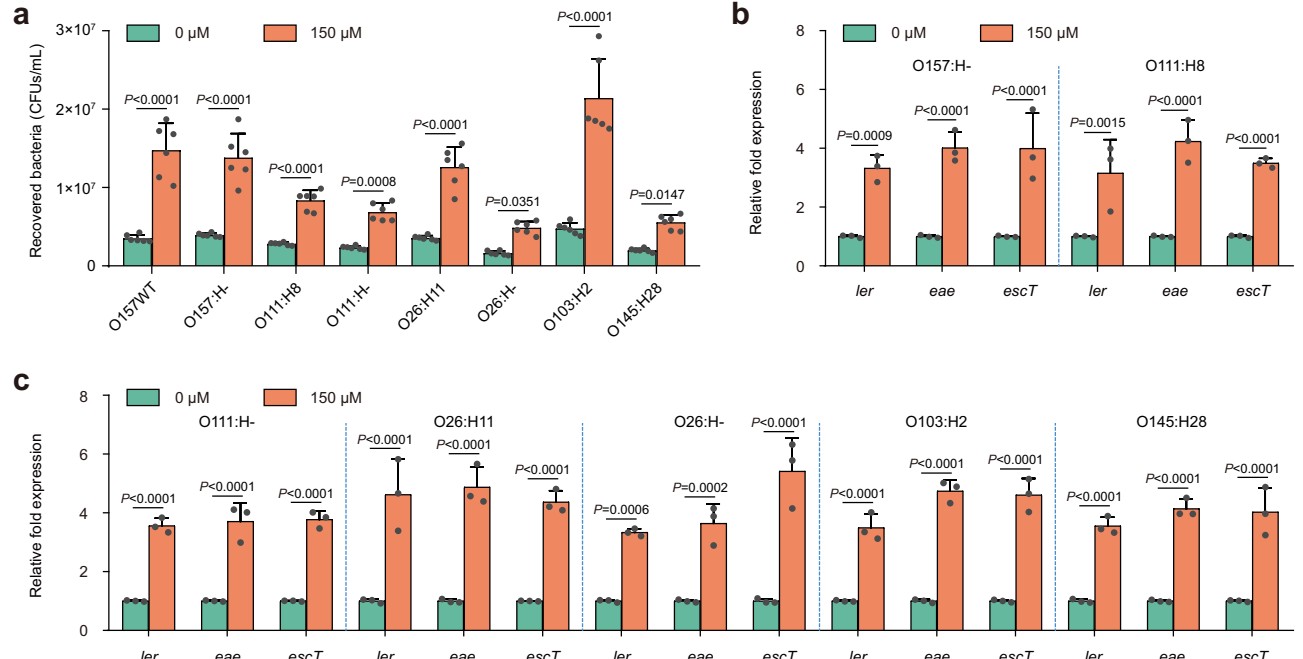

**Fig. 7 | The effect of L-malate on bacterial adherence and LEE gene expression is conserved in EHEC. a** Adherence of various EHEC strains to HeLa cells in DMEM supplemented with 0 or 150 μM L-malate. **b, c** qRT–PCR to determine LEE gene expression changes in various EHEC strains grown in DMEM supplemented with 0 or 150 μM L-malate. In (**a**), data are presented as the mean ± SD of six independent biological replicates (*n* = 6). In (**b, c**), data are presented as the mean ± SD of three independent biological replicates (*n* = 3). Statistical significance was assessed via two-way ANOVA followed by Sidak's post hoc test. Source data are included in Source Data file.

mechanism used by a range of EHEC strains to control bacterial virulence.

## Discussion

The human gastrointestinal tract is a complex environment populated by a dense and diverse microbiota that plays important roles in the physiology of the intestine[6,42]. In addition to providing nutrients and vitamins, the microbiota has an important role as a protective barrier against enteric pathogens, commonly referred to as colonization resistance[6,42]. However, these enteric pathogens employ multiple systems to sense and respond to different microbiota-derived signals, as well as host-derived signals and nutrients, to coordinate the expression of their virulence traits and adjust their metabolism to ensure successful competition for limited nutrients and a colonization niche[7,8]. In this study, we found that EHEC O157:H7 utilizes host- and microbiota-derived L-malate both as a nutrient source to benefit growth and as a signal molecule to activate virulence gene expression during infection. A model for this L-malate signaling regulatory pathway in EHEC O157:H7 is proposed (Fig. 8). Briefly, when EHEC O157:H7 enters the large intestine where the level of L-malate is high, the expression of genes involved in anaerobic L-malate transport and metabolism, including *dcuB*, *fumB* and *frdABCD*, which encode the anaerobic L-malate transporters, the fumarase FumB, and fumarate reductase, is activated by the DcuSR TCS. EHEC O157:H7 then efficiently imports L-malate and subsequently converts it to fumarate to fuel anaerobic fumarate respiration, thereby promoting bacterial growth and colonization in vivo. Moreover, EHEC O157:H7 also responds to L-malate in the large intestine as a signaling molecule to activate virulence gene expression. The DcuSR TCS senses L-malate and transduces the signal to the master virulence regulator Ler, which in turn activates LEE gene expression to promote EHEC O157:H7 T3SS-dependent adherence to large intestinal epithelial cells. Regardless, acquisition of the ability to simultaneously utilize L-malate in the large intestine as a nutrient source that provides a competitive advantage for colonization over the

microbial flora and as a signaling molecule to activate virulence gene expression is important for the evolution of EHEC O157:H7 as a successful invader of the human large intestine.

What is the source of the L-malate in the intestine of infant rabbits? The L-malate levels in the colon of infant rabbits fed a normal diet (rabbit breast milk) and those fed an L-malate-free diet (sterilized water containing 10% lactose) were similar (Figs. 1c, 2d), ruling out the existence of a dietary source of L-malate. Intestinal colonization by the Δ*dcuABC* and the Δ*frdABCD* mutants in germ-free infant rabbits fed an L-malate-free diet was still significantly attenuated compared with colonization by wild-type EHEC O157:H7 (Supplementary Fig. 5a), indicating that L-malate is partly derived from the host. We also found that treatment of infant rabbits with antibiotics significantly reduced the concentration of L-malate in the colon (Fig. 1c). However, it was unclear from this result whether this reduction occurred due to depletion of the microbiota or because antibiotic treatment altered the mitochondrial metabolism of the host cells. Previous studies revealed that the microbial metabolite short-chain fatty acids (SCFAs), primarily butyrate, provide an energetic source for colon epithelial cells and can affect mitochondrial metabolism in many ways[43,44]. The decrease in microbial SCFAs caused by antibiotic treatment will cause the oxygen metabolism of intestinal epithelial cells to change from oxidative phosphorylation to glycolysis, which will increase the oxygenation of epithelial cells, promote the diffusion of oxygen into the intestinal lumen and disturb the anaerobic environment[45,46]. In addition, the decreased mitochondrial viability of epithelial cells caused by the decrease in butyrate-producing bacteria will lead to the upregulation of the expression of *Nos2* (encoding inducible nitric oxide synthase iNOS), which ultimately leads to the increased production of nitric oxide[45]. To investigate whether antibiotic treatment and the resulting changes in SCFAs affect the production and release of L-malate by host cells, the L-malate levels were determined in HeLa cell supernatants and lysates in the presence of a cocktail of four antibiotics (ampicillin, neomycin, metronidazole, and vancomycin;

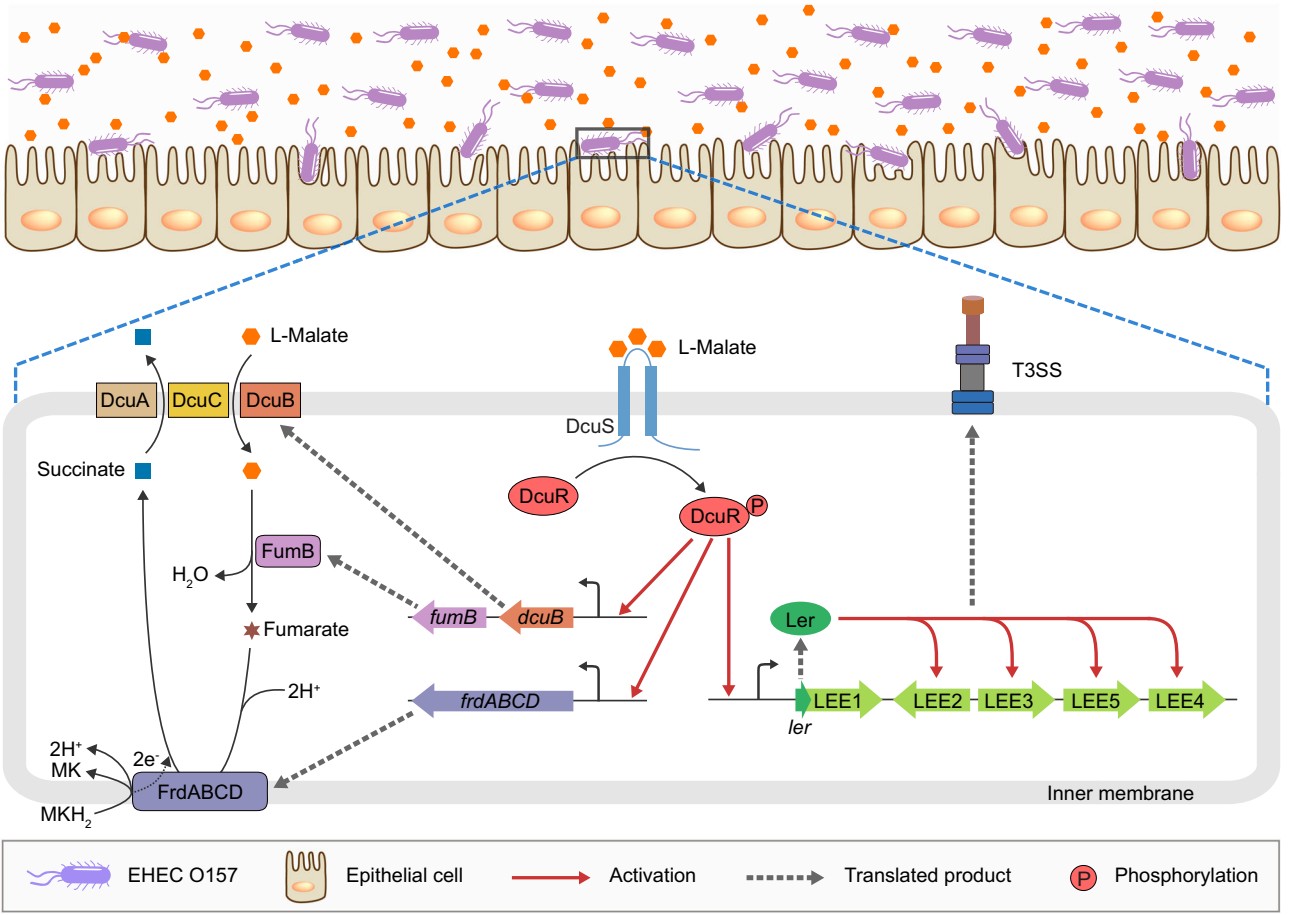

**Fig. 8 | Model for the DcuSR-mediated L-malate signaling regulatory pathway in EHEC O157:H7.** When EHEC O157:H7 enters the human large intestine, where L-malate levels are high, the TCS sensor kinase DcuS responds to the high L-malate signal and undergoes autophosphorylation, after which the phosphoryl group is transferred between DcuS and its cognate DNA-binding transcriptional regulator DcuR. On the one hand, phosphorylated DcuR (DcuR-P) directly activates the expression of genes involved in anaerobic L-malate transport and metabolism, including *dcuB*, *fumB* and *frdABCD*, which encode anaerobic L-malate transporters, fumarase FumB, and fumarate reductase. These highly expressed genes ensure that

EHEC O157:H7 efficiently imports L-malate and subsequently converts it into fumarate to fuel anaerobic fumarate respiration, thereby promoting bacterial growth and colonization in the large intestine. On the other hand, DcuR-P directly activates the expression of *ler* (encoding the master regulator of LEE genes) by binding the *ler* promoter, after which Ler activates other LEE genes to promote EHEC O157:H7 T3SS-dependent adherence to the epithelial cells of the large intestine. MK menaquinone, $MKH_2$ menaquinol, LEE locus of enterocyte efface-ment, T3SS type III secretion system.

---

100 μg/mL each) or 10 mM butyrate. The results showed that the L-malate levels in cell supernatants and lysates were not affected by the presence of antibiotics and butyrate (Supplementary Fig. 5b, c), indicating that the decreased L-malate concentration in the colonic contents of infant rabbits following antibiotic treatment (Fig. 1c) is due to depletion of the microbiota rather than altered production and release of L-malate by host cells. Thus, L-malate in the intestine of infant rabbits originates from both the host and the microbiota.

To better distinguish the role of L-malate as a signal molecule in promoting bacterial adherence from that of L-malate as a nutrient source to benefit bacterial growth in vivo, infant rabbit competition experiments were performed to evaluate the intestinal colonization capacity of Δ*ler* (defective in T3SS-dependent adherence) in the infant rabbit intestine. The results showed that the Δ*ler* mutant exhibited an approximately 75-fold attenuation of intestinal colonization compared with wild-type EHEC O157:H7 (mean CI = 0.014; Supplementary Fig. 6), and this severe attenuation is similar to that of the Δ*dcuABC* triple mutant (defective in L-malate transport; Fig. 2a and Supplementary Fig. 6). These results indicated that L-malate acts as a signal molecule in promoting bacterial adherence and as a nutrient source to benefit growth and is equally critical for successful colonization of EHEC O157:H7 in the high-L-malate large intestine.

To further investigate whether EHEC O157:H7 utilizes other common C4-dicarboxylates (such as L-aspartate, fumarate, or succinate) as essential nutrients and signaling molecules for intestinal colonization, the levels of L-aspartate, fumarate and succinate in the colon of infant rabbits were further quantified using the corresponding assay kits (Sigma: MAK095, MAK060, MAK335). The colons of infant rabbits contained intermediate levels of succinate (~75 nmol/g luminal contents) and low levels of L-aspartate (~10 nmol/g luminal contents), whereas fumarate was nearly absent (~0.25 nmol/g luminal contents) (Supplementary Fig. 7a). Succinate is the reduction end product of fumarate respiration, and it cannot serve as a substrate for further anaerobic respiration during bacterial growth and colonization in vivo[30,47]. In addition, the concentrations of free fumarate and L-aspartate in the colon might be too low to support EHEC O157:H7 growth. Consistent with this speculation, deletion of *aspA*, which is unable to convert L-aspartate to fumarate, had no obvious effects on colonization by EHEC O157:H7 in the intestines of infant rabbits (Supplementary Fig. 7b). Therefore, EHEC O157:H7 mainly utilizes L-malate rather than other C4-dicarboxylates (aspartate, fumarate and succinate) for fumarate respiration during colonization of the large intestine of infant rabbits. qRT–PCR analysis showed that the expression of the LEE genes was not affected by the presence of 750 μM

succinate, 100 µM L-aspartate or 2.5 µM fumarate (corresponding to the concentrations in the colon of infant rabbits) (Supplementary Fig. 7c). Therefore, L-aspartate, fumarate and succinate are unlikely to be sensed as intestinal signals by EHEC O157:H7 when activating virulence gene expression and promoting bacterial adherence in the large intestine. It is more likely that high L-malate levels are the only signal that the DcuSR TCS responds to in the large intestine when positively regulating EHEC O157:H7 adherence and LEE gene expression.

In *E. coli* and related bacteria, L-malate is converted to fumarate and then used in fumarate respiration[25,26]. In this pathway, fumarate is used as an electron acceptor for anaerobic respiration and reduced to succinate[25,26]. Succinate, as the reduction end product of fumarate respiration, cannot be oxidized because of the nonfunctional TCA cycle and is therefore excreted by the DcuABC antiporters[25,26]. Some enteric pathogens have evolved mechanisms to utilize succinate as a signaling molecule that induces their own virulence and survival in vivo. For example, *Salmonella* Typhimurium senses succinate accumulation in macrophages to promote antimicrobial resistance and type III secretion[48]. Active transport of succinate through DcuB is necessary for *Salmonella* Typhimurium virulence and survival within macrophages and mice[48]. Similarly, EHEC O157:H7 senses high levels of succinate via an unknown mechanism and induces the catabolite repressor/activator protein Cra, which in turn activates T3SS expression and AE lesion formation[49]. Therefore, in addition to acting as a nutrient source to benefit growth and as a signal molecule to directly activate LEE gene expression, L-malate also contributes to EHEC O157:H7 virulence activation by generating another virulence regulatory signal, succinate, through fumarate respiration.

The current knowledge of the regulatory role of the DcuSR TCS in bacteria is limited to the control of the synthesis of fumarate reductase (encoded by *frdABCD*), the fumarase FumB (encoded by *fumB*) and the anaerobic fumarate-succinate antiporter DcuB (encoded by *dcuB*) in response to C4-dicarboxylates such as fumarate, L-malate, D-tartrate and aspartate[33,50]. Here, we found that the DcuSR TCS senses L-malate and directly activates *ler* gene expression by binding phosphorylated DcuR to the LEE1 promoter, after which Ler activates the expression of other LEE genes to promote EHEC O157:H7 adherence. Therefore, DcuSR TCS also plays a crucial regulatory role in bacterial virulence regulation, and the results of this study significantly expand our understanding of the regulatory function of the DcuSR TCS.

Our bacterial adherence assays, FAS assays and qRT–PCR analysis showed that deletion of *dcuS* or *dcuR* in EHEC O157:H7 resulted in significant reductions in adherence to HeLa cells and Caco-2 cells (Fig. 5a), AE lesion formation (quantified by the percentage of HeLa cells infected and the number of pedestals per infected cell; Fig. 5b), and LEE gene expression (Fig. 5c), even in the absence of L-malate. These results indicated that the DcuSR TCS retains basic (unstimulated) activity and function in the absence of external C4-dicarboxylate stimulation. Consistent with these results, previous studies also found that DcuS is able to autophosphorylate and transfer phosphoryl groups to DcuR in the absence of C4-dicarboxylates (fumarate, succinate and malate)[34,51]. We hypothesize that in addition to C4-dicarboxylates, the DcuSR TCS could also be directly or indirectly activated by other signal molecules; however, testing this hypothesis requires further investigation.

Previous studies found that *E. coli* is able to use L-malate for growth under aerobic and anaerobic conditions[25,52]. However, we found here that the growth of EHEC O157:H7 under aerobic conditions at 37 °C in DMEM was not affected by the presence of 150 µM or 500 µM L-malate (Fig. 4c). DMEM contains a large amount of glucose (~25 mM), which has been demonstrated to be the optimal carbon source for bacterial growth. Furthermore, glucose suppresses secondary metabolic pathways such as L-malate utilization through carbon catabolite repression. To support rapid growth, *E. coli* often undergoes overflow metabolism in the presence of high glucose levels,

meaning that *E. coli* uses fermentation instead of the more efficient respiration pathway to generate energy, despite the availability of oxygen. In contrast, L-malate was added to DMEM at relatively low concentrations (150 µM and 500 µM) and acted primarily as an electron acceptor under anaerobic conditions. Therefore, the absence of growth differences is not surprising and can be reasonably explained by the above facts.

Mass spectrometry (MS) is an attractive possibility for carbohydrate detection, especially because MS is a fast and high-throughput analysis method[53,54]. Part of the difficulty in monosaccharide structure determination by MS is the numerous isomers for a given chemical composition[55,56]. In particular, most of these isomers are stereoisomers, meaning that they are only different in terms of the orientation of their chemical bonds; this makes structural differentiation challenging[55,56]. In this study, untargeted metabolomics analysis was performed using a liquid chromatography–tandem mass spectrometry (LC/MS/MS) approach. Metabolites were identified based on accurate mass (m/z, ± 25 ppm), retention time and MS/MS spectra against an in-house metabolite database, which contains chemical standards and a manually curated compound list generated from running purified compound standards through the experimental platforms. Almost all isomers can be distinguished by these three criteria. For example, although mannose-1-phosphate, fructose-1-phosphate and galactose-1-phosphate have the same m/z value, different retention times (479.99 s, 306.52 s and 461.42 s, respectively) can be used to distinguish these isomers. However, untargeted metabolomics can only provide relative quantitative analysis; thus, other quantitative approaches, such as targeted metabolomics and biochemical methods, are also needed to accurately quantify the levels of metabolites identified by untargeted metabolomics.

Peptidoglycan (PG) is a major component of the bacterial cell wall[57]. PG is built up of glycan strands of alternating N-acetylglucosamine (GlcNAc) and N-acetylmuramic acid (MurNAc) residues that are connected by short peptides to form a mesh-like polymer. Most bacteria, including *E. coli*, contain D-Ala-D-Ala at positions four and five of the stem peptide[57]. It is estimated that up to 50% of the PG is shed and degraded per bacterial generation[58]. During PG degradation, various peptidases cleave the amide bonds within the stem peptide to form amino acids and small peptides (such D-alanyl-D-alanine), which can be reused by *E. coli* as direct precursors for PG and protein biosynthesis and as carbon and nitrogen sources for metabolism[58,59]. Our metabolomics results showed that D-alanyl-D-alanine was present at greater abundance in the colon of infant rabbits than in the ileum (Fig. 1b and Supplementary Data 1). However, other components of PG have not been identified by our untargeted metabolomics analysis. Determining whether EHEC O157:H7 utilizes D-alanyl-D-alanine as a nutrient and/or a virulence gene regulatory signal for colonization in the host large intestine requires further investigation.

In addition to L-malate and D-alanyl-D-alanine, several other metabolites commonly found in the mammalian intestine, including L-arabinose, lithocholic acid, deoxycholic acid, myristic acid, and nicotinate, were also significantly more abundant in the large intestine than in the small intestine (Fig. 1b and Supplementary Data 1). L-Arabinose is one of the most abundant components released by complete hydrolysis of nonstarch polysaccharides in the mammalian intestine[60]. L-Arabinose can improve intestinal health by manipulating the composition of the gut microbiota[61]. A recent study found that L-arabinose inhibits Shiga toxin type 2-converting bacteriophage induction in EHEC O157:H7[62]. Lithocholic acid and deoxycholic acid are secondary bile acids that are prevalent in the cecum and colon of mice and humans[63,64]. In the colon, unconjugated bile acids are transformed into lithocholic acid and deoxycholic acid via 7α-dehydroxylation, a highly efficient process performed by resident genera, including *Lactobacillus*, *Bifidobacterium*, *Clostridium*, *Bacteroides* and *Enterococcus*[63].

Lithocholic acid was reported to impair the separation of growing vancomycin-resistant *Enterococcus* diplococci, causing the formation of long chains and increased biofilm formation[64]. In *Salmonella*, deoxycholic acid directly represses the expression of invasion-related genes, including *sipC*, *sopB*, and *hilD*[65]. Myristic acid, a long-chain saturated fatty acid, is one of the most abundant fatty acids in milk fat (above 10%)[66]. Transcriptomic analysis showed that myristic acid repressed the expression of several biofilm-related genes (*csgAB*, *fimH* and *flhD*) in EHEC O157:H7[67]. Additionally, myristic acid reduced the virulence of EHEC O157:H7 in a nematode infection model and exhibited minimal cytotoxicity[67]. Nicotinate, also known as vitamin B3, is a water-soluble vitamin that can be made from tryptophan in mammals as well as from intestinal bacteria[68]. In *Bordetella pertussis*, nicotinate inhibits the kinase activity of BvgS, preventing transcription of virulence genes[69]. We recently found that EHEC O157:H7 senses microbiota-derived nicotinamide (structural analog of nicotinate) to increase its virulence and colonization in the large intestine[70]. Whether these large intestine-abundant metabolites are utilized by EHEC O157:H7 as nutrients and/or virulence-regulating signaling molecules during colonization in the large intestine needs to be further investigated in future studies.

Additionally, we also found that several L-amino acids, including L-lysine, L-arginine, L-threonine and L-glutamine, were present at greater abundance in the small intestine (Fig. 1b and Supplementary Data 1). This is understandable since dietary proteins are mainly hydrolyzed by proteases and peptides in the small intestine to generate free amino acids[71]. These digestive amino acids are absorbed by enterocytes of the small intestine via different transporters[71]. However, our metabolomics analysis fails to identify other amino acids produced by protein degradation, which may be due to the different amounts of different amino acids in the diet and/or the different absorption efficiency of different amino acids by intestinal epithelial cells.

Rabbits and mice are the two most common animal species used to study EHEC-mediated disease. Mice must undergo artificial manipulations, such as antibiotic treatment, to eliminate the normal intestinal flora and facilitate EHEC colonization[36,37]. Furthermore, mice did not develop diarrhea, colitis, or AE lesions following EHEC O157:H7 inoculation[36,37]. Three-day-old infant rabbits provide a more readily available animal model without the requirement for additional treatments, wherein EHEC O157:H7 caused diarrhea, colonic inflammation and death[38,39]. Histological inflammatory changes were seen mainly in the mid- and distal colon with increased apoptosis (individual cell death) in the surface epithelium, increased mitotic activity in the crypts, mucin depletion, and a mild to moderate infiltrate of neutrophils in the lamina propria and epithelium[72]. The histological changes were much less severe in the proximal colon, cecum, and gut-associated lymphoid tissue and were minimal in the small intestine[72]. Inflammatory diarrhea also developed consistently in 11-day-old rabbits, although the EHEC-related disease was less severe even with a higher inoculum dose[72]. In contrast, EHEC O157:H7 did not produce inflammatory diarrhea in older rabbits (20 days old) and weaned young rabbits (approximately 6 weeks old)[72,73]. The reason for this age-dependent susceptibility is not clear but may be related to differences in microbiota composition and intestinal immune system development across age groups. We speculate that the susceptibility of 3-day-old rabbits to EHEC O157:H7 may be due to the mature microbiome and intestinal immune system that have not yet developed in this animal model.

Although L-malate promoted the adherence of all 7 tested EHEC strains to host cells, the fold change in bacterial adherence varied significantly among the different EHEC strains (Fig. 7a). Furthermore, even in the absence of L-malate, there was considerable variation in their adherence capabilities (Fig. 7a). These strains belong to 7 different serotypes and were isolated from different geographical locations (Supplementary Data 2), which may partly explain the large differences in their adherence phenotypes. We also found that although L-malate-promoted bacterial adherence varied greatly among different EHEC strains, the fold changes in the expression of LEE genes (*ler*, *eae*, and *escT*) in the presence of 0 and 150 µM L-malate were negligible among the different EHEC strains (Fig. 7b, c). These results suggested that, in addition to LEE genes, other virulence factors may also be involved in L-malate-promoted EHEC adherence to host cells; verification of this hypothesis requires further investigation.

EHEC O157:H7 infection has emerged as an important new zoonosis, giving rise to serious public health concerns in North America, Europe, and, increasingly, other areas of the world[40]. The Center for Disease Control and Prevention (CDC) estimated that EHEC O157:H7 causes approximately 73,000 cases of illness and 60 deaths annually in the United States[74]. The annual cost of illness due to EHEC O157:H7 infections was estimated to be more than 400 million dollars based on various causes, including lost productivity, medical care, and premature deaths[74]. Although several strategies have been developed, including the use of antibiotics and vaccination[75,76], EHEC O157:H7 infection cannot be treated effectively by current therapies. The use of antibiotics is also contraindicated because antibiotics are either ineffective, cause severe dysbiosis of the intestinal microbiota, or trigger serious complications due to antibiotic-induced release of Shiga toxin[74]. Considering that deletion of *dcuS* or *dcuR* significantly decreased EHEC O157:H7 adherence and virulence gene expression both in vitro and in vivo, these virulence-related genes may be used as potential targets for the development of new therapeutics for the prevention and control of EHEC O157:H7 infection.

## Methods

### Ethics Statement

All animal experiments were performed according to the standards established by the Guide for the Care and Use of Laboratory Animals published by the Institute of Laboratory Animal Resources of the National Research Council (United States). The experimental protocols were approved by the Institutional Animal Care Committee at Nankai University, Tianjin, China (protocol number: SYDWLL-000168). Every effort was made to minimize animal suffering and to reduce the number of animals used.

### Bacterial strains, plasmids and growth conditions

The bacterial strains and plasmids used in this study are listed in Supplementary Data 2. Mutant strains and FLAG-tagged strains were generated using the λ Red recombinase system[77], and all strains were verified via PCR amplification and sequencing. Complementation strains of the mutants were generated by cloning appropriate genes and their promoter regions into the low-copy plasmid pACYC184, and the resulting constructs were then electroporated into the corresponding mutant strains. Strains for protein overexpression and purification were constructed by cloning genes of interest into the pET28a expression plasmid, and the resulting constructs were then electroporated into the *E. coli* BL21 strain. All the resulting constructs were verified by DNA sequencing. Primers for all experiments are summarized in Supplementary Data 3. The bacterial strains were grown at 37 °C in Luria−Bertani (LB) broth or DMEM under either anaerobic or aerobic conditions. When necessary, antibiotics were added at the following final concentrations: nalidixic acid, 50 µg/mL; ampicillin, 100 µg/mL; chloramphenicol, 15 µg/mL; and kanamycin, 50 µg/mL.

### Bacterial adherence assays

Bacterial adherence assays were performed according to a previously described method[15,78] with slight modification. HeLa and Caco-2 cells were purchased from the Shanghai Institute of Biochemistry and Cell Biology of Chinese Academy of Science (Shanghai, China). HeLa and Caco-2 cells were grown at 37 °C in 5% $CO_2$ until confluent and then subcultured in a 6-well plate for 24 h. Before infection, cells were

washed with prewarmed phosphate-buffered saline (PBS) three times, and fresh DMEM without antibiotics and fetal bovine serum (FBS) was added. The cell monolayers were then infected with exponential-phase bacteria grown in DMEM at an MOI of 100:1. Three hours after incubation with cells, unattached bacteria were removed by washing the wells six times with PBS. The cells were then disrupted with 0.1% SDS, and lysate dilutions were plated on LB agar plates. Attachment efficiency was calculated as colony-forming units (CFUs) per milliliter.

## FAS assays

FAS assays were performed as described previously[79,80] with some modifications. Briefly, HeLa cells were grown on coverslips in 6-well culture plates containing DMEM supplemented with 10% FBS and incubated at 37 °C in 5% $CO_2$ overnight to approximately 85% confluence. The wells were washed with PBS, and fresh DMEM supplemented with 10% FBS was added. Bacterial cultures were grown aerobically overnight and then diluted 1:100 to infect HeLa cells for 6 h at 37 °C in 5% $CO_2$. The coverslips were washed, fixed with formaldehyde and then permeabilized with 0.2% Triton X. Fluorescein isothiocyanate (FITC)-labeled phalloidin was used to visualize actin accumulation, and propidium iodide was used to visualize host nuclei and bacteria. The coverslips were then mounted on slides and visualized with a Zeiss LSM800 confocal microscope using a 63× objective and 488-nm and 561-nm excitation lasers.

## Quantitative RT–PCR (qRT–PCR)

qRT–PCR reaction was performed in the QuantStudio™ 5 Real-Time PCR System (Applied Biosystems). For in vitro experiments, cultures were grown to mid-exponential phase (optical density 0.8 at 600 nm) in DMEM under microaerobic conditions. For in vivo experiments, colonic contents were collected from infected infant rabbits on day 2 after EHEC O157:H7 infection and flash frozen in liquid nitrogen. The samples were homogenized with liquid nitrogen. Total RNA was extracted using TRIzol® LS Reagent (Invitrogen: 15596018) and treated with RNase-Free DNase I (Qiagen: 79254) to eliminate genomic DNA contamination. First-strand cDNA was synthesized using the Prime-Script 1st Strand cDNA Synthesis Kit (Takara: D6110A). qRT–PCR was carried out in a total volume of 20 μL in a 96-well optical reaction plate containing 10 μL of Power SYBR Green PCR Master Mix (Applied Biosystems: 4367659), 1 μL of cDNA, and two gene-specific primers, each at a final concentration of 0.3 mM. rpoA was used as a reference control for sample standardization, and the relative differences in gene expression were calculated using the $2^{-\Delta\Delta Ct}$ method[81].

## Western blotting assays

Overnight bacterial cultures were diluted to an optical density at 600 nm of 1.0. Whole-cell lysates were separated via 12% SDS-polyacrylamide gel electrophoresis and transferred to polyvinylidene difluoride membranes. The membranes were treated with TBST (Tris-buffered saline with Tween 20) containing 5% nonfat milk for 1 h at room temperature to block nonspecific binding. Subsequently, the membranes were probed with anti-FLAG (1:2,500 dilution, Sigma: F1804) or anti-DnaK (1:5,000 dilution, Abcam: ab69617) primary antibody for 1 h, followed by a goat anti-mouse immunoglobulin G conjugated to horseradish peroxidase (1:5000 dilution, Abcam: ab205719) secondary antibody for 1 h. Blots were visualized on a chemiluminescence detection system (GE Healthcare) following an enzyme immunoassay using ECL enhanced chemiluminescence reagent. DnaK was used as a loading control, and proteins were quantified by densitometry and normalized to DnaK. Western blotting bands were quantified using ImageJ software.

## EMSAs and competition assays

The promoter regions of LEE1, LEE2/3, LEE4 and LEE5 and the rpoS fragment (negative control) were amplified using genomic DNA of the wild-type EHEC O157:H7 EDL933 strain as a template and the corresponding FAM-labeled primers shown in Supplementary Data 3. In each reaction, 50 nM each FAM-labeled DNA fragment was incubated with various concentrations of purified proteins (0–4 μM) in a 20 μL solution containing the binding buffer [10 mM Tris-HCl (pH 7.5), 80 mM NaCl, 0.1 mM EDTA, 0.2 mM dithiothreitol, and 5% v/v glycerol]. For competition assays, various concentrations of unlabeled DNA fragments (0.5–5 μM) were added. The reaction mixtures were incubated at 37 °C for 30 min and then separated using a native 10% polyacrylamide gel in 0.5× Tris-borate-EDTA. The DNA fragments and DNA–protein complexes were visualized using an Amersham Imager 600 (GE Healthcare).

## Chromatin Immunoprecipitation-quantitative PCR (ChIP–qPCR)

An inducible expression vector (pTRC99a) carrying 3×FLAG-tagged dcuR was constructed and transformed into the ΔdcuR mutant. Bacterial cultures were grown to mid-exponential phase until the optical density at 600 nm reached 0.4, and protein expression was induced with 1 mM IPTG at 37 °C for 30 min. ChIP was performed based on established methods as reported previously[82] with some modification. Formaldehyde was added to bacterial cultures at a final concentration of 1% for crosslinking of protein to DNA, and the mixture was then incubated at room temperature for 25 min. Reactions were quenched with 0.5 M glycine (final concentration). The cross-linked cells were sonicated to generate DNA fragments of approximately 500 bp following washing with ice-cold TBS three times. Cell debris was removed, and protein–DNA complexes were enriched using an anti-FLAG antibody (1:2,500 dilution, Sigma: F1804) and protein A magnetic beads (Invitrogen: 10002D). As a negative control, chromatin immunoprecipitation was performed on the other aliquot without the addition of antibodies. To measure the enrichment of candidate gene promoters in immunoprecipitated DNA, the relative abundance was determined by quantitative PCR with Power SYBR Green PCR Master Mix (Applied Biosystems: 4367659). The relative enrichment of DNA regions of interest was calculated by the $2^{-\Delta\Delta Ct}$ method[81].

## Dye primer-based DNase I footprinting assay

The promoter region of LEE1 was amplified using EHEC O157:H7 EDL933 as a template and the 6-FAM-labeled forward primer (with 6-FAM modification at the 5′ end) and reverse primer. The 6-FAM-labeled probe (40 nM) was then incubated with 2.5 μM DcuR in binding buffer, after which the protein–DNA mixture was partially digested with 0.05 units of DNase I for 5 min at 25 °C. The reaction was terminated by heating the mixture at 80 °C for 10 min, and the product was purified using the Qiaquick PCR Purification Kit (Qiagen: 28104). Control samples were prepared with bovine serum albumin (BSA) instead of the DcuR protein. The genotype samples were analyzed using the 3730 Genetic Analyzer and viewed with Peak Scanner v1.0 software (Applied Biosystems).

## Untargeted metabolomics analysis

Three-day-old female New Zealand White (NZW) rabbits were necropsied, and the mid-ileal contents and mid-colonic contents were harvested and resuspended in water at 0.5 g/mL. The samples were vortexed vigorously and then centrifuged at 2500 × g for 5 min. The supernatant was collected. Untargeted metabolomics relative-quantitative analysis was performed by Shanghai Applied Protein Technology Co., Ltd. (Shanghai, China) using a liquid chromatography–tandem mass spectrometry (LC/MS/MS) approach. The raw MS data (wiff.scan files) were converted to MzXML files and processed for feature detection, retention time correction and alignment. Compound identification for metabolites was performed by comparing the accurate m/z value (<25 ppm) with the MS/MS spectra with an in-house database established with available authentic standards. The significantly different metabolites were determined based on the

combination of a statistically significant threshold of variable influence on projection (VIP) values obtained from running the OPLS-DA model and the two-tailed Student's t test (p value) on the raw data. The metabolites with VIP values larger than 1.0 and p values less than 0.05 were considered statistically significant. The intestinal content preparation for untargeted metabolomics analysis was performed in four independent experiments. The colonic contents from infant rabbits were also collected as described above (concentrated if necessary) and used for quantitation of the levels of L-malate, L-aspartate, fumarate and succinate using the corresponding assay kits (Sigma: MAK067, MAK095, MAK060, MAK335). To eliminate the gut microflora for the L-malate measurements, infant rabbits were orally administered a cocktail of four antibiotics, ampicillin, neomycin, metronidazole, and vancomycin, via oral gavage for 2 d (5 mg each antibiotic per rabbit per day).

## Animal experiments

Three-day-old female New Zealand White (NZW) rabbits were purchased from Beijing Huabukang Biological Technology Co., Ltd. (Beijing, China) and housed under standard laboratory conditions with a 12 h light/dark cycle, at a temperature of $24 \pm 2$ °C, and a relative humidity of $50 \pm 5\%$. Infant rabbits were fed a L-malate-free diet (50 µL of sterilized water containing 10% lactose) or L-malate-rich diet (50 µL of sterilized water containing 10% lactose and 10 mM L-malate) at 6 h intervals. To compare the colonization ability of EHEC O157:H7 in the small and large intestine, three-day-old female NZW rabbits were orally infected by pipette feeding of 100 µL of PBS containing $10^9$ CFUs of exponential-phase wild-type EHEC O157:H7 (Nal$^R$). At various times after infection (1 d, 2 d, 3 d, 4 d, 5 d, 6 d and 7 d), the infant rabbits were anesthetized with isoflurane and euthanized by means of cervical dislocation. Midileal and midcolonic intestinal tissues were collected from individual infant rabbits, weighed, and homogenized in sterile PBS. Serial dilutions of the homogenate were plated onto LB agar plates containing 30 µg mL$^{-1}$ nalidixic acid. To determine the effect of the L-malate status of the infant rabbit intestinal tract on colonization by EHEC O157:H7, three-day-old female NZW rabbits were intragastrically administered 50 µL of sterilized water containing 10% lactose (L-malate-free diet) or 50 µL of sterilized water containing 10% lactose and 10 mM L-malate (L-malate-rich diet) at 6 h intervals for 2 d. Ten infant rabbits in each group were intragastrically inoculated ($10^9$ CFUs) with wild-type EHEC O157:H7. The rabbits were necropsied 4 d postinoculation, and the contents and tissues of the mid-colon were collected from individual infant rabbits, weighed and homogenized in sterile PBS. The homogenates were subsequently diluted and plated on LB agar containing appropriate antibiotics. The colonization efficiency in vivo was calculated as the number of CFUs per gram of intestinal tissue or contents. In the competition experiments, groups of infant rabbits were intragastrically inoculated (combined final concentration of $10^9$ CFUs) with a 1:1 ratio of wild-type EHEC O157:H7 and its mutant or complemented strains. Competitive indexes were calculated using the relative abundance of each strain in the colonic contents, corrected by the ratio in the inoculum.

## Statistical analysis

Statistical analyses were performed using GraphPad Prism 8 software. For normal data with equal variance, Student's t test was used when comparing two groups. For normal data with equal variance, one-way or two-way ANOVA was used when comparing more than two groups. For nonnormal data with unequal variance (e.g., infant rabbit colonization experiments), the Mann–Whitney rank-sum test was used for comparisons between two groups, and the Kruskal–Wallis test with Dunn's post hoc test was used for comparisons between more than two groups. $P$ values < 0.05 were considered statistically significant. Statistical details for all tests performed can be found in the figure legends.

## Reporting summary

Further information on research design is available in the Nature Portfolio Reporting Summary linked to this article.

## Data availability

The metabolomic data generated in this study have been deposited in the MetaboLights database under accession code MTBLS8754 [www.ebi.ac.uk/metabolights/MTBLS8754]. All other data associated with this study are available in the main text and supplementary materials. Source data are provided with this paper.

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

## Acknowledgements

This work was supported by the National Science Foundation of China (NSFC) Program under Grant no. 32270191 (to B.Y.) and 31800125 (to B.Y.), the Fundamental Research Funds for the Central Universities under Grant no. 63233172 (to B.Y.) and the Natural Science Foundation of Tianjin under Grant no. 20JCQNJC0197 (to B.Y.).

## Author contributions

B.Y., B.L. and L.J. designed the research; Y.L., H.S., J.Y. and C.K. performed the research; Y.L., H.S., J.Y. and C.K. collected the data; B.Y., B.L., L.J. and Y.L. analyzed the data; and B.Y., B.L. and L.J. wrote the paper.

## Competing interests

The authors declare no competing interests.
