## [Peer Review File · Nature Communications]

Enterohaemorrhagic *E. coli* utilizes host- and microbiota-derived L-malate as a signaling molecule for intestinal colonizationEditorial Note: This manuscript has been previously reviewed at another journal that is not operating a transparent peer review scheme. This document only contains reviewer comments and rebuttal letters for versions considered at *Nature Communications*.

REVIEWERS' COMMENTS

Reviewer #1 (Remarks to the Author):

The authors have adequately addressed my comments about a previous version of the manuscript.

Reviewer #2 (Remarks to the Author):

Liu et al. have diligently addressed reviewers' comments, made substantial revisions to their manuscript, and conducted additional experiments to enhance the quality of their work. In recent years, fumarate respiration and C4-dicarboxylates have garnered increasing attention within the research community due to emerging evidence linking them to virulence and colonization in murine models. Liu et al.'s study takes this knowledge further by shedding light on their relevance in the context of Enterohemorrhagic *Escherichia coli* (EHEC) infection in infant rabbit models. The research provides valuable insights into the role of C4-dicarboxylates, such as L-malate, L-aspartate, fumarate, and succinate, which apparently exhibit varying degrees of importance in different organisms and animal models.

I am confident that this research will strongly resonate with the readership of *Nature Communications*.

Minor points:

- In line 112, it appears that the authors intended to refer to the micromolar range.
- In line 417, it should be noted that intestinal cells transition from β -oxidation of butyrate, involving oxidative phosphorylation (a form of mitochondrial aerobic respiration), to lactate fermentation. This involves glycolysis and lactate fermentation, which do not utilize oxygen as an electron acceptor. Therefore, the term "aerobic glycolysis" is not accurate.
- Between lines 514 and 521, it is crucial to mention that glucose, through carbon catabolite repression, suppresses secondary metabolic pathways like L-malate utilization. *Escherichia coli* often undergoes overflow metabolism in the presence of high glucose levels, meaning it ferments even when oxygen is available. Furthermore, as the authors themselves noted, the L-malate concentration was relatively low. Thus, the absence of growth differences is unsurprising. The primary advantage of C4-dicarboxylates like L-malate and fumarate lies in their role as electron acceptors under anaerobic conditions.

Reviewers Comments:

Reviewer #1 (Remarks to the Author):

The authors have adequately addressed my comments about a previous version of the manuscript.

Reviewer #2 (Remarks to the Author):

Liu et al. have diligently addressed reviewers' comments, made substantial revisions to their manuscript, and conducted additional experiments to enhance the quality of their work. In recent years, fumarate respiration and C4-dicarboxylates have garnered increasing attention within the research community due to emerging evidence linking them to virulence and colonization in murine models. Liu et al.'s study takes this knowledge further by shedding light on their relevance in the context of Enterohemorrhagic Escherichia coli (EHEC) infection in infant rabbit models. The research provides valuable insights into the role of C4-dicarboxylates, such as L-malate, L-aspartate, fumarate, and succinate, which apparently exhibit varying degrees of importance in different organisms and animal models.

I am confident that this research will strongly resonate with the readership of Nature Communications.

Minor points:

1. In line 112, it appears that the authors intended to refer to the micromolar range.

Response:

Thank you for the comment. The term “millimolar range” has now been modified to “micromolar range” in the revised manuscript (pg. 5, line 99).

2. In line 417, it should be noted that intestinal cells transition from β -oxidation of butyrate, involving oxidative phosphorylation (a form of mitochondrial aerobic respiration), to lactate fermentation. This involves glycolysis and lactate fermentation, which do not utilize oxygen as an electron acceptor. Therefore, the term "aerobic glycolysis" is not accurate.

Response:

Thank you for the comment. The term “aerobic glycolysis” has now been modified to “glycolysis” in the revised manuscript (pg. 19, line 397).

3. Between lines 514 and 521, it is crucial to mention that glucose, through carbon catabolite repression, suppresses secondary metabolic pathways like L-malate utilization. *Escherichia coli* often undergoes overflow metabolism in the presence of high glucose levels, meaning it ferments even when oxygen is available. Furthermore, as the authors themselves noted, the L-malate concentration was relatively low. Thus,

the absence of growth differences is unsurprising. The primary advantage of C4-dicarboxylates like L-malate and fumarate lies in their role as electron acceptors under anaerobic conditions.

Response:

Thank you for the comment.

Previous studies found that *E. coli* is able to use L-malate for growth under aerobic and anaerobic conditions. However, we found here that the growth of EHEC O157:H7 under aerobic conditions at 37°C in DMEM was not affected by the presence of 150 µM or 500 µM L-malate (Fig. 4c). DMEM contains a large amount of glucose (~ 25 mM), which has been demonstrated to be the optimal carbon source for bacterial growth. Furthermore, glucose suppresses secondary metabolic pathways such as L-malate utilization through carbon catabolite repression. To support rapid growth, *E. coli* often undergoes overflow metabolism in the presence of high glucose levels, meaning that *E. coli* uses fermentation instead of the more efficient respiration pathway to generate energy, despite the availability of oxygen. In contrast, L-malate was added to DMEM at relatively low concentrations (150 µM and 500 µM) and acted primarily as an electron acceptor under anaerobic conditions. Therefore, the absence of growth differences is not surprising and can be reasonably explained by the above facts. The relevant information has been added to the Discussion section of the revised manuscript (pg. 23, lines 489-494; pg. 24, lines 495-501).